# Rapid Diagnostic Test to Detect and Discriminate Infectious Hematopoietic Necrosis Virus (IHNV) Genogroups U and M to Aid Management of Pacific Northwest Salmonid Populations

**DOI:** 10.3390/ani12141761

**Published:** 2022-07-09

**Authors:** William N. Batts, Tony R. Capps, Lisa M. Crosson, Rachel L. Powers, Rachel Breyta, Maureen K. Purcell

**Affiliations:** 1U.S. Geological Survey, Western Fisheries Research Center, Seattle, WA 98115, USA; bbatts@usgs.gov (W.N.B.); rpowers@usgs.gov (R.L.P.); 2Washington Department of Fish and Wildlife, Olympia, WA 98501, USA; tony.capps@dfw.wa.gov (T.R.C.); lisa.crosson@dfw.wa.gov (L.M.C.); 3School of Aquatic and Fisheries Sciences, University of Washington, Seattle, WA 98195, USA; rbjmax@uw.edu

**Keywords:** infectious hematopoietic necrosis virus, IHNV, diagnostic accuracy, sensitivity, specificity, strain discrimination, genogroup, salmonid, Columbia River Basin

## Abstract

**Simple Summary:**

Infectious hematopoietic necrosis virus (IHNV) can cause severe disease and mortality in salmon and trout. Two major groups of the virus are found in the North American Columbia River Basin and Washington Coast, designated U and M. The U and M groups pose different risks depending on the fish host species. Here, we report the development of a new diagnostic test that can both detect the presence of IHNV and rapidly discriminate between the U and M groups. The new assay proved sensitive and specific when applied to free ranging, returning adult Pacific salmon.

**Abstract:**

Infectious hematopoietic necrosis virus (IHNV) is an acute pathogen of salmonids in North America, Europe, and Asia that is phylogenetically classified into five major virus genogroups (U, M, L, E, and J). The geographic range of the U and M genogroup isolates overlap in the North American Columbia River Basin and Washington Coast region, where these genogroups pose different risks depending on the species of Pacific salmon (*Oncorhynchus* spp.). For certain management decisions, there is a need to both test for IHNV presence and rapidly determine the genogroup. Herein, we report the development and validation of a U/M multiplex reverse transcription, real-time PCR (RT-rPCR) assay targeting the IHNV nucleocapsid (N) protein gene. The new U/M RT-rPCR is a rapid, sensitive, and repeatable assay capable of specifically discriminating between North American U and M genogroup IHNV isolates. However, one M genogroup isolate obtained from commercially cultured Idaho rainbow trout (*O. mykiss*) showed reduced sensitivity with the RT-rPCR test, suggesting caution may be warranted before applying RT-rPCR as the sole surveillance test in areas associated with the Idaho trout industry. The new U/M assay had high diagnostic sensitivity (DSe > 94%) and specificity (DSp > 97%) in free-ranging adult Pacific salmon, when assessed relative to cell culture, the widely accepted reference standard, as well as the previously validated universal N RT-rPCR test. The high diagnostic performance of the new U/M assay indicates the test is suitable for surveillance, diagnosis, and confirmation of IHNV in Pacific salmon from the Pacific Northwest regions where the U and M genogroups overlap.

## 1. Introduction

Infectious hematopoietic necrosis virus (IHNV; *Novirhabdovirus salmonid*) is a highly lethal rhabdovirus of wild and cultured salmonid fishes [1,2]. Phylogenetic analysis has identified five major virus genogroups of IHNV worldwide, designated U, M, L, E, and J [3]. Genogroups U, M, and L are all present in western North America and correspond to the upper, middle, and lower portions of the IHNV geographic range [4]. The U genogroup ranges from Alaska through Oregon and inland throughout the Columbia River Basin (CRB) [5], the M genogroup is found throughout the CRB and sporadically on the Washington Coast [6], and the L is found in southern Oregon and California [7]. The European E genogroup IHNV strains are ancestrally derived from the M genogroup, and the Asian J genogroup ancestrally derived from the U genogroup [3]. The genus *Novirhabdovirus* is within the family *Rhabdoviridae* and includes IHNV, viral hemorrhagic septicemia virus (*N. piscine*, VHSV), hirame rhabdovirus (*N. hirame*, HIRRV), and snakehead rhabdovirus (*N. snakehead*, SHRV) [3,8].

Given the severity of disease caused by IHNV in economically and ecologically significant salmonids, detection of the virus is reportable to the World Organization of Animal Health (OIE) and many state, tribal, and provincial North American natural resource agencies [9,10]. Within the Columbia and Snake Rivers, both U and M genogroups of IHNV are likely to occur in salmonid populations each year but not all watersheds have both U and M detections [11,12]. Strains belonging to the M genogroup of IHNV are highly lethal to rainbow trout and steelhead *Oncorhynchus mykiss* [13,14]. Strains belonging to the U genogroup are highly lethal to sockeye salmon *O. nerka* [13,15,16]. Genogroups of IHNV can be further divided into subgroups. In the CRB, the UC subgroup is commonly found in both Chinook salmon *O. tshawytscha* and steelhead and is less frequently associated with juvenile mortality, unlike the UP subgroup that primarily infects and causes juvenile disease in sockeye salmon outside this basin [17]. Four main subgroups exist within the M genogroup with the MD subgroup commonly infecting CRB steelhead and causing juvenile mortality [6,18]. Since UC and MD strains are common in the CRB but pose different risks to various species, managers often desire the ability to discriminate between these two strains. This has taken on a greater need with plans to re-establish salmonids in historical ranges where fish passage was blocked by dam operations [19]. Some blocked reaches have native trout species that have never been exposed to the more recently evolved M genogroup of IHNV [11]. Typically distinguishing between U and M strains requires DNA sequencing and phylogenetic analysis. A rapid test to discriminate between strains would provide fish health managers the ability to quickly ascertain the risk when moving salmonids between watersheds and/or to evaluate potential disease risks to host populations.

While immunological, histological, and conventional PCR tests for IHNV are recommended by the OIE for presumptive diagnosis of clinically affected fish and/or confirmatory diagnosis, only cell culture and a validated reverse transcriptase real-time PCR (RT-rPCR) are recommended for all purposes including IHNV surveillance of apparently healthy fish [20]. The OIE recommended test is a rapid, sensitive, and validated RT-rPCR test that targets a conserved region of the nucleocapsid (N) gene and allows universal detection of known strains (hereafter referred to as the N Uni RT-rPCR assay) [21,22]. While other RT-rPCR assays for IHNV have been reported previously [23,24,25], these are not currently recommended by the OIE. None of the previously developed RT-rPCR assays discriminate specific genogroups. The goal of our study was to develop and validate an RT-rPCR assay with high sensitivity to detect IHNV and high specificity to discriminate between the known North American U and M genogroup strains. To fit our purpose, the assay must be able to identify known U and M IHNV isolates, not cross-react with other viral species, and reliably detect viral loads typically found in infected salmon populations. Here, we report the development of the new U/M RT-rPCR assay, as well as the analytical and diagnostic validation of this new test in comparison to the OIE recommended cell culture (our reference standard) and N Uni RT-rPCR tests. The new U/M multiplex RT-rPCR assay provides a simple, fast, sensitive, and specific diagnostic tool that produces reliable results suitable for aiding management decisions.

## 2. Materials and Methods

### 2.1. Assay Development

A thorough search of U and M IHNV sequences, as well as other *Novirhabdovirus* sequences (Appendix A), revealed only one region suitable for the locked nucleic acid (LNA) multiplex probe design to distinguish between U and M strains. Three nucleotide mismatches distinguished the U and M strains in the candidate locked nucleic acid (LNA) probe binding site (Table 1). The U/M RT-rPCR primer and probe sites were adjacent but slightly upstream of the N Uni RT-rPCR primer and probe sites [21]. For discrimination of U and M genogroup viruses within the same reaction, the U LNA probe was labeled with HEX 5′ reporter dye and the M LNA probe was labeled with 6FAM 5′ reporter dye; both probes had a 3′ Iowa Black quencher (Integrated DNA Technologies (IDT), Inc., Coralville, IA, USA). An artificial positive control (APC) plasmid [21,26] and a gBlock gene fragment based on the same sequence as the APC were designed to encode the RT-rPCR target region (Appendix A). The U/M assay artificial constructs, as well as the constructs previously developed for the N Uni assay [21], were used as assay positive controls and for the analytical validation stage. 

### 2.2. RT-rPCR Reaction Conditions

Extracted RNA from cell culture grown IHNV or laboratory infected steelhead was used for initial testing of the U/M RT-rPCR assay. Culture of IHNV in EPC, harvesting of culture supernatant and RNA extraction was performed as described [21]. A juvenile steelhead (Dworshak National Fish Hatchery origin) was reared to 1.1 g in specific pathogen-free water, artificially exposed simultaneously to UC (strain DW10) and MD (strains QTS07) to generate a co-infected fish, as described by [27], sampled at 3 days post-exposure and RNA from whole fry was extracted [28]. High Capacity cDNA Synthesis Kit (Life Technologies, Inc., Waltham, MA, USA) was used for RNA reverse transcription, as described by [21]. Assay parameters were as previously described for the N Uni RT-rPCR [21] as these conditions performed suitably for the new U/M RT-rPCR. Briefly, the ViiA-7 Real-Time PCR System or the QuantStudio 6 Flex Real Time PCR System (Life Technologies) were used with cycling conditions of 50 °C for 2 min, 95 °C for 10 min, followed by 40 cycles of 95 °C for 15 s and 60 °C for 1 min. Each 12 µL total reaction contains: 7 µL of TaqMan Universal PCR Master Mix (Life Technologies, Inc.) with 900 nM forward primer, 900 nM reverse primer, 200 nM of IHNV U probe, 200 nM of IHNV M probe, and 5 µL of cDNA (1:5 diluted in water). Negative template controls and positive standards were included on each plate run. 

### 2.3. Analytical Validation

To evaluate analytical specificity (ASp), the U/M and N Uni RT-rPCR assays were tested for cross-reactivity to five fish rhabdovirus RNA samples: VHSV (Genogroups Ia, IVa, and IVb), HIRRV, and SHRV (Table 2). A total of 58 IHNV isolates representing U (21), M (31), L (2), E (2), and J (2) genogroups were tested to compare IHNV detection using multiplex U/M assay to the validated N Uni RT-rPCR assay (Table 2). An increased number of U and M representatives from the CRB was tested to ensure the reliability of the RT-rPCR assays for this region. 

To evaluate PCR efficiency and analytical sensitivity (ASe), serial 10-fold dilutions of the APC plasmid DNA, gBlock DNA, and infected steelhead cDNA were subjected to the U/M and N Uni RT-rPCR assays. The quality of standard curves was evaluated by the coefficient of determination (r^2^), slope, and y-intercept. For infected steelhead, mean cycle threshold (C_T_) values were converted to mean log_10_ IHNV copies per reaction as previously described [21].

Within-run repeatability of the U/M RT-rPCR and N Uni assay was estimated by performing five replicate wells (in the same run) of U (UP–BLK94) and M (MD–Qts-07) isolates representing three different viral load levels, higher (~22 CT), medium (~25 CT), and lower (~29 CT). To measure the between-run repeatability of the U/M assay, the three samples representing different viral load levels were tested in six separate runs on different days. Coefficient of variation (CV) was calculated as standard deviation/mean. 

### 2.4. Diagnostic Validation

Sample collection, cell culture, and RT-rPCR testing was performed by trained personnel of the WDFW Fish Health program. Returning adult Chinook salmon (*n* = 50), sockeye salmon (*n* = 43), and steelhead (*n* = 9) were collected as part of WDFW’s routine hatchery surveillance program from January 2019 to December 2020; these samples represent a random subset that were selected for paired analysis by cell culture and the U/M RT-rPCR. Kidney/spleen tissues were collected using established fish health inspection methods and freshly collected tissue was subjected to cell culture using the EPC cell line (www.atcc.org, accession #CRL-2872) (Accessed on 8 July 2022) [10]. Individual tissues from fish were pooled for all analyses; samples that did not contain 5 fish were excluded from our analysis for consistency purposes. Duplicate tissues samples were taken for molecular analysis and stored at −80 °C for <6 months until used. RNA was extracted with RNeasy Mini-Kit (Qiagen, Germantown, MD, USA) following the manufacturer’s instructions for tissue and RT-rPCR reactions were performed as described above. Samples from six WA facilities were included: Baker Lake (*n* = 30), Cedar River (*n* = 13), Cowlitz (*n* = 12), Kalama Falls (*n* = 12), Marblemount (*n* = 6), and Soos Creek (*n* = 29). The frequency of IHNV at these facilities has historically varied; for instance, adult sockeye salmon from Baker Lake and Cedar River are frequently infected with IHNV while Soos Creek Chinook salmon and steelhead do not have a history of IHNV. All fish pools were subjected to cell culture and the U/M RT-rPCR (N = 102), while only a subset of the pools was subjected to the N Uni RT-rPCR (N = 63). The criterion for considering an rRT-PCR test result positive was reproducible amplification in technical replicates with a mean C_T_ value < 37; mean C_T_ values > 37.0 but <40.0 were considered a suspect result; and no evidence of amplification was considered a test negative result. Viruses isolated by cell culture were identified by conventional PCR and DNA sequencing of the glycoprotein (G) gene using the OIE recommended procedure [20]. Diagnostic parameters were estimated using online tools available from Ausvet (https://epitools.ausvet.com.au; accessed on 13 May 2022). This publication adheres to the ‘Standards for Reporting of Animal Diagnostic Accuracy Studies for Aquatic Species’ (STRADAS-aquatic) [29] and all study data are publicly available [30]. 

## 3. Results

### 3.1. Assay Development and Fit for Purpose

An alignment of available genome sequences of U and M IHNV genogroups and related rhabdoviruses (Appendix A) identified a 79 nt region in the N gene ideal for designing primers and probes. The U/M RT-rPCR assay included a variable probe region bounded by conserved sites for primer amplification. Our goal was to develop a test that could discriminate between the U and M genogroups. As such, the sole criterion of probe selection was consistent differences between U and M sequences, regardless of what other IHNV genogroups (L, J, and E) might contain at the polymorphic sites. Initial testing of cell culture grown isolates of known IHNV genogroups indicated that the new multiplex U/M RT-rPCR assay detected and discriminated the U and M genogroups demonstrating a good fit for purpose. Optimization testing indicated that previously described conditions for the N Uni RT-rPCR were optimal. Both APC plasmid and gBlock constructs performed suitably as controls, providing linear amplification over 7 orders of magnitude with coefficient of determination values (R^2^) consistently >0.99 (Table 3). 

### 3.2. Analytical Validation

A panel of 21 U genogroup IHNV isolates were all correctly detected by the U probe without any cross-reaction by M probe and 31 M genogroup IHNV isolates were all correctly detected by the M probe without any cross-reaction by the U probe (Table 2). There was no detectable amplification for the most closely related fish rhabdovirus isolates VHSV, HIRRV, or SHRV (Table 2). The U/M assay performed similarly to the previously validated N Uni assay, with both tests showing 100% ASp. As expected from sequence similarity, the U probe also detected L and J genogroup IHNV and the M probe detected E genogroup IHNV.

The C_T_ for all IHNV isolates ranged from 14.4 to 25.2 by the previously validated N Uni assay, 13.1–24.3 for the U probe, and 13.6–28.0 for the M probe (Table 1). For the M IHNV genogroup, isolates Hg508 and 17-054 had a higher than expected C_T_ value for the N Uni assay, while isolate 17-073 had a higher than expected C_T_ value for the U/M assay. Although both assays still detected the isolates, the results suggested a reduced sensitivity for these three isolates. DNA sequencing revealed a single nucleotide polymorphism (SNP) in the N Uni probe site for Hg508, an SNP in the N Uni Rev primer site for 17-054, and an SNP in the U/M assay Rev primer for 17-073.

The ASe of the U/M and N Uni RT-rPCR assays was estimated by comparing dilution series of the APC plasmid, gBlock constructs, and co-infected fish tissue cDNA. Both assays showed similar slopes and good PCR efficiency (E) across all sample types, ranging from 92.7 to 101.4% (Table 3). For both the N Uni and U/M assays, the limit of detection (LOD) was <10 copies/reaction for the control constructs (APC and gBlock) and <30 copies/reaction for the infected fish tissue samples. The N Uni and U/M assays showed linear amplification across a large dynamic range (7 log range tested for the APC and gBlock standards; 5–6 log range tested for infected fish RNA). 

The CV of three U and M IHNV cDNA samples with higher (~22 CT), medium (~25 CT), and lower (~29 CT) copy numbers tested by the N Uni and U/M assays with 4–5 technical replicates indicated high within-run repeatability (CVs ranging from 0.6 to 1.4% depending on sample copy number and assay). These three samples were tested across 6 independent runs for the U/M assay and showed high between-run repeatability (CV ranging from 1.1 to 2.0%). 

### 3.3. Diagnostic Validation

Diagnostic performance of the U/M RT-rPCR for screening returning adult salmon populations was evaluated by comparing the U/M assay results to two established IHNV diagnostic tests, cell culture, and N Uni RT-rPCR (Table 4). The U/M assay had high agreement with both cell culture (97.1%) and N Uni (95.2%), with the Kappa (Κ) statistic suggesting almost perfect agreement between tests. No matter the comparative test, the U/M RT-rPCR assay had high diagnostic sensitivity (DSe = 0.94) and diagnostic specificity (DSp > 0.97). The C_T_ values obtained from the U/M RT-rPCR ranged from 18.7 to 38.0 and N Uni RT-rPCR tests ranged from 18.9 to 39.2; the C_T_ values were highly correlated (R^2^ = 0.99). Among the 102 samples that were subjected to both cell culture and the U/M RT-rPCR, there were only three discrepant samples. Two of the discrepant samples tested positive by cell culture and negative by the U/M assay and one sample tested positive by U/M assay (36.7 C_T_) but negative by cell culture. All viruses isolated as part of this dataset were U genogroup and 100% of the samples that tested positive by the U/M RT-rPCR assay were detected with only the U specific probe (USGS data release [30]). We also observed high DSe and DSp for the N Uni RT-rPCR assay, using cell culture as the reference standard, further validating the suitability of the N Uni RT-rPCR for surveillance of adult Pacific salmon. 

## 4. Discussion

The analytical sensitivity, specificity, and repeatability of the new U/M RT-rPCR was comparable to the OIE recommended N Uni RT-PCR assay [21]. Furthermore, the U/M assay had high diagnostic performance when used for surveillance of asymptomatic adult salmon and steelhead, when compared to either the reference standard cell culture or the N-Uni RT-rPCR. The U/M assay accurately discriminated between genogroups when applied to 52 diverse U and M virus isolates. This accuracy was corroborated with correct genogroup identification of 34 new U viruses isolated during the 2019–2020 adult salmon surveillance effort. While the U/M assay detected and correctly identified U/M isolates, it did not detect the closely related rhabdoviruses VHSV, HIRRV, and SHRV. As predicted based on the design of the U/M assay, the U probe also detected the L and J genogroups from California and Asia and the M probe detected the E genogroup viruses from Europe. However, there were no L, J, or E strain detections in the Pacific Northwest region, except for a rare detection of L strain on the southern Oregon coast. Thus, the new U/M RT-rPCR assay achieved the intended purpose to both detect and discriminate between North American U and M IHNV genogroup strains. Given the high DSe (94%) and DSp (>95%), the assay appears suitable for surveillance, diagnosis, and confirmation of IHNV in Pacific Northwest regions where the U and M genogroups overlap. 

Both the N Uni and U/M RT-rPCR assays target a conserved region of the N gene, while IHNV strain classification is based on sequencing of a diverse 303 nucleotide region of the G gene. The N gene is transcribed at higher levels than the G and other viral genes [31], making assays targeting the N gene potentially more sensitive to active infections. Previous RT-rPCR assays targeting the G gene [24,25] had difficulty detecting a wide range of IHNV strains. In the present study, we noted an isolate of the MB subgroup (strain Hg508; genotype mG296-M; isolated in 2014) and an isolate of the MD subgroup (strain 17-054; mG107M; isolated in 1999) had an SNP in either the Rev primer or probe of the N Uni RT-rPCR. An isolate of the MC subgroup (strain 17-073; genotype mG352-M; 2012) had an SNP in U/M assay Rev primer. All three of these isolates were obtained from Idaho rainbow trout farms, an industry and region associated with unusually high rates of IHNV genetic diversity [32,33]. In Germany, reduced sensitivity has been reported for the N Uni RT-rPCR when detecting E genogroup isolates associated with the rainbow trout industry due to the presence of nucleotide substitutions in the probe region [34]. A redesigned probe, implemented as a one-step PCR protocol, showed improved sensitivity for a limited number of German E isolates of IHNV tested in the study. The diagnostic application of the redesigned Hoferer et al. 2019 RT-rPCR assay outside Germany is difficult to assess because the study only evaluated three older North American strains (SRCV LII-1966, WRAC MA-1982, and RB76 UP-1976). Additional work is needed to empirically assess the ASe and ASp of the redesigned assay in contemporary U, M, L, or J isolates, as well as E isolates from outside Germany. The findings of E and M genogroup IHNV variants with reduced RT-rPCR assay sensitivity associated with the German and Idaho rainbow trout industries may indicate a need for targeted and ongoing diagnostic validation of molecular IHNV tests in these farmed populations, as well as the continued paired use of cell culture when IHNV is suspected. 

The evaluation of the new U/M assay was performed across two laboratories (USGS and WDFW). The assay performed well in both laboratories using RNA derived from both cell culture supernatant and kidney/spleen tissues. The assays were highly repeatable in both laboratories showing similarly low within and between run variability (<2.7% at low copy numbers; see data release [30]). However, both the U/M and N Uni RT-rPCR assays performed poorly when applied to RNA samples derived from ovarian fluid. Inclusion of an internal positive control (IPC) in the reaction indicated that the poor detection was not caused by PCR inhibition. Rather, the poor results were due to low RNA recovery using the RNeasy MiniKit tissue protocol. Pilot testing showed that improved RNA yield from ovarian fluid samples was obtained with TriReagent-LS or TriReagent (Sigma-Aldrich, Burlington, MA, USA), Direct-zol RNA extraction kit (Zymo Research, Irvine, CA, USA), and QiaAmp Viral RNA Mini Kit (Qiagen). We did not include the ovarian fluid sample results in the diagnostic validation phase of this study because the samples were subjected to several rounds of freezing and thawing. However, these issues emphasize the importance of optimizing the RNA extraction method for ovarian fluid. Our laboratories applied a two-step RT-rPCR protocol for both the N Uni and U/M assays in this study. However, the N Uni RT-rPCR had equivalent analytical performance when modified to a one-step protocol [22,30]. We did not fully validate the one-step protocol, but preliminary testing of the assay using a commercial one-step master mix (TaqMan Fast Virus 1-Step Master Mix) yielded similar results as the two-step. Thus, the assay could be modified to a one-step protocol with an in-house comparability assessment, as described by the OIE [35]. 

The threshold value at which a diagnostic test result is considered positive or negative should be clearly defined [36]. The criteria we used herein to consider a sample positive was reproducible technical replicates and mean C_T_ < 37 (~50 copies/reaction), which was slightly greater than the reliable LOD (10–30 copies/reaction). Samples with non-reproducible replication or mean C_T_ values > 37–40 were considered suspect, while samples with no evidence of amplification by 40 cycles were considered negative. The 37 C_T_ cut-off was in a similar range to other published RT-rPCR studies [35,37] and was selected to reduce the potential of management actions based on false positive or equivocal results. Particularly since samples with C_T_ values > 37 may be difficult to confirm with secondary tests (e.g., cell culture or conventional RT-PCR [21]). Nonetheless, samples that yield a suspect test result can still be subjected to additional technical replicates or secondary conventional RT-PCR testing to clarify results. The U/M assay was designed for use in the Pacific Northwest where IHNV is widespread and the concerns of a false positive vs. false negative result are more equally balanced. For certain purposes, a more conservative approach may be warranted (e.g., amplification at any C_T_) when the concerns of a false negative test result outweigh the concerns for a false positive result. 

Genogroups U and M IHNV co-circulate throughout the salmonid populations of the CRB and Washington Coast but the different genogroups may present different risks depending on the region and species. Cell culture isolation followed by conventional PCR/DNA sequencing can require several weeks to detect and identify the IHNV genogroup. In certain situations, there is a need for rapid discrimination of the U and M strains. For instance, the emergence of the M genogroup in Washington Coast steelhead populations from 2007 to 2011 was met with more aggressive management actions, which may have helped to prevent the establishment of this subgroup on the coast [6,38]. Our primary motivation for developing the U/M assay was to support Tribal efforts to restore anadromous salmon to their former ranges that have been historically blocked by hydropower operations [11]. These programs will require translocation of adult salmon to river reaches above dams, but some of these reaches may be naïve to the M genogroup. The M genogroup is hypothesized to have evolved from the U genogroup via a host jump and specialization in rainbow trout (ca. 1970s), becoming established in the CRB in the early 1980s [5]. As an example, fish passage in the Upper Columbia River has been blocked since construction of the Grand Coulee Dam in 1938 and reaches above the dam have naïve resident *O. mykiss* species that may be highly susceptible to the M genogroup. Thus, the new U/M assay can help mitigate risks of IHNV by serving as a useful tool to rapidly screen adults proposed for translocation. However, it should be noted that additional study is needed to fully estimate the diagnostic accuracy of the IHNV RT-rPCR assays when applied to non-lethally collected fin or gill tissues. Previous studies using cell culture indicate that fin tissue could be a useful non-lethal tissue for IHNV surveillance of adult salmonids [39]. 

## 5. Conclusions

The new U/M RT-rPCR is a rapid, sensitive, and specific method to both detect and discriminate between North American U and M genogroup IHNV. The U/M assay had high diagnostic sensitivity and specificity in free-ranging adult Pacific salmon as compared to cell culture and the validated N Uni RT-rPCR. The high diagnostic sensitivity and specificity indicates that the assay is suitable for surveillance, diagnosis, and confirmation of IHNV in Pacific Northwest regions where the U and M genogroups overlap. However, additional diagnostic validation may be warranted before applying any RT-rPCR test as the sole surveillance tool in regions associated with the Idaho rainbow trout industry due to increased frequency of IHNV genetic variants in this population.

## Figures and Tables

**Table 1 animals-12-01761-t001:** Primer and probe configuration for the U/M RT-rPCR assay. The “+” indicates location of locked nucleic acids (LNA) in probes, bold and underlined bases show three sites of discrimination between U and M genogroups.

Primer/Probe Name	Sequence (5′–3′)
IHNV U/M 739F	ACCAAGGCTATCTATGGGATCATTCTCAT
IHNV U/M 817R	TGGCGCACAGTGCCTTG
U Probe (785-795)	HEX-CC+A+C+CGCCGCT-IABkFQ/LNA
M Probe (783-795)	6-FAM-AGCC+A+T+CGC+TGCC-IABkFQ/LNA

**Table 2 animals-12-01761-t002:** Evaluation of analytical specificity (ASp) of the U/M multiplex reverse transcriptase real-time PCR (RT-rPCR) assay compared to the N Universal (N Uni) RT-rPCR. RNA extracted from viral isolates, including infectious hematopoietic necrosis virus (IHNV), viral hemorrhagic septicemia virus (VHSV), hirame rhabdovirus (HIRRV), and snakehead rhabdovirus (SHRV). Bolded values with asterisk indicate a higher-than-expected C_T_, suggesting reduced assay sensitivity for that strain. C_T_: cycle threshold; bd: below detection. N uni: N-gene universal IHNV probe, U/M multi: multiplex IHNV probes detecting either the U or M genogroup.

Viral Species	Major Genogroup	Sub-Group	Isolate Name	Geographic Area	Year	Mean C_T_ N Uni	Mean C_T_ U Probe	Mean C_T_ M Probe
IHNV	U	P	Auke77	AK, USA	1977	18.2	16.5	bd
	U	P	GF77	AK, USA	1977	17.0	13.9	bd
	U	P	BC4814	BC, Canada	1992	14.8	13.1	bd
	U	P	Blk94	WA, USA	1994	17.4	15.6	bd
	U	P	RU1	Russia	2000	18.3	18.2	bd
	U	P	RU9	Russia	2001	15.4	13.6	bd
	U	P	BC203	BC, Canada	2002	17.5	15.9	bd
	U	P	CdR12	WA, USA	2012	17.0	13.9	bd
	U	P	MM15	WA, USA	2015	17.5	13.5	bd
	U	C	RB1	OR, USA	1975	17.3	15.6	bd
	U	C	RB5	OR, USA	2005	16.5	15.5	bd
	U	C	LYF05	WA, USA	2005	17.4	16.4	bd
	U	C	KSK08	WA, USA	2008	20.5	19.5	bd
	U	C	LYF09	WA, USA	2009	16.3	15.4	bd
	U	C	LYC09	ID, USA	2009	18.5	17.5	bd
	U	C	LNFH10	WA, USA	2010	18.1	16.6	bd
	U	C	DW10	ID, USA	2010	17.3	15.5	bd
	U	C	17-020	WA, USA	2015	17.8	18.2	bd
	U	C	17-019	WA, USA	2015	18.6	17.6	bd
	U	C	17-031	WA, USA	2016	25.2	24.3	bd
	U	Asia	Shiz82	Japan	1982	21.3	19.6	bd
	M	N	LR80	WA, USA	1980	17.4	bd	17.1
	M	A	WRAC	ID, USA	1982	18.1	bd	17.7
	M	B	220-90	ID, USA	1990	16.2	bd	16.0
	M	B	17-067	ID, USA	2008	21.6	bd	18.7
	M	B	Hg508	ID, USA	2014	**20.0 ***	bd	15.8
	M	C	C30	ID, USA	1991	16.2	bd	15.5
	M	C	17-051	ID, USA	1999	18.0	bd	17.3
	M	C	17-073	ID, USA	2012	19.8	bd	**28.0 ***
	M	D	SK81	WA, USA	1981	14.4	bd	13.6
	M	D	KK89	ID, USA	1989	17.3	bd	16.3
	M	D	SK94	WA, USA	1994	16.1	bd	15.3
	M	D	Mer95	WA, USA	1995	16.7	bd	16.6
	M	D	Qts97	WA, USA	1997	16.1	bd	15.3
	M	D	17-054	ID, USA	1999	**24.2 ***	bd	17.3
	M	D	17-058	ID, USA	1999	21.7	bd	21.0
	M	D	SK02	WA, USA	2002	16.3	bd	15.5
	M	D	Tan02	WA, USA	2002	16.8	bd	17.0
	M	D	SK04	WA, USA	2004	15.1	bd	14.5
	M	D	CW06	WA, USA	2006	15.7	bd	14.9
	M	D	Qts07	WA, USA	2007	16.5	bd	17.7
	M	D	DW09	ID, USA	2009	19.6	bd	18.9
	M	D	Qts10	WA, USA	2010	16.3	bd	15.2
	M	D	17-074	ID, USA	2013	16.2	bd	16.0
	M	D	17-075	ID, USA	2013	18.7	bd	18.4
	M	D	17-081	ID, USA	2014	17.5	bd	16.5
	M	D	Hg511	ID, USA	2014	15.5	bd	14.6
	M	D	17-087	ID, USA	2015	19.6	bd	18.6
	M	D	17-098	ID, USA	2015	20.2	bd	20.0
	M	D	17-100	ID, USA	2015	17.8	bd	17.5
	M	D	17-014	WA, USA	2015	20.4	bd	20.2
	M	D	17-023	WA, USA	2015	17.3	bd	17.1
	L	I	C18	CA, USA	1991	20.5	18.9	bd
	L	II	FR0031	CA, USA	2000	22.3	21.1	bd
	E	-	32/87	France	1987	21.2	bd	20.9
	E	-	55/94	Austria	1994	19.4	bd	18.9
	J	-	ChAb76	Japan	1976	18.0	16.4	bd
	J	-	RtUi02	Korea	2002	19.0	14.7	bd
VHSV	I	A	DK3592B	Denmark	1986	bd	bd	bd
	IV	A	Makah	WA, USA	1988	bd	bd	bd
	IV	B	MI03	MI, USA	2003	bd	bd	bd
HIRRV	-	-	8401H	Japan	1984	bd	bd	bd
SHRV	-	-	-	Thailand	1986	bd	bd	bd

**Table 3 animals-12-01761-t003:** Analytical sensitivity (ASe) of the N Uni and U/M RT-rPCR assays determined by analyzing 10-fold serially diluted artificial positive control (APC) plasmid DNA, gBlock DNA, or cDNA from fish infected with infectious hematopoietic necrosis virus (IHNV). Cycle threshold (C_T_) shown for duplicate technical replicates of each dilution (APC plasmid or gBlock) or quadruplicate technical replicates of each dilution (infected fish tissues). The limit of detection (LOD) indicates the dilution where all technical replicates detected amplification. E: PCR efficiency.

Template Source		N Uni			U/M Assay U Probe			U/M Assay M Probe	
	Slope	E	LOD	Slope	E	LOD	Slope	E	LOD
APC ^1^	−3.51	92.7%	<10	−3.38	97.6%	<10	−3.36	98.4%	<10
gBlock ^1^	−3.49	93.4%	<10	−3.45	94.9%	<10	−3.44	95.3%	<10
Fish ^2^	−3.50	93.1%	<20	−3.33	99.7%	<30	−3.29	101.4%	<10

^1^ APC plasmid and gBlock standards showed linear amplification with both assays across a 7 log range; R^2^ > 0.99. ^2^ RNA from fish artificially co-infected with U/M strains was evaluated across a 5–6 log dilution range and both assays showed linear amplification; R^2^ > 0.96.

**Table 4 animals-12-01761-t004:** Diagnostic performance of the U/M and N Uni RT-rPCRs. The U/M assay was compared to two previously validated diagnostic tests for IHNV, viral isolation by culture in EPC cells (reference standard), and the N Uni RT-rPCR. The N Uni RT-rPCR was compared to only viral isolation. Samples were taken from Washington Department of Fish and Wildlife’s routine surveillance program for returning adult sockeye salmon, Chinook salmon, and steelhead. All samples represent 5 fish pools of kidney/spleen. A total of 102 pools were evaluated by both cell culture and the U/M RT-rPCR, while 63 pools were evaluated by both the U/M and N Uni RT-rPCRs.

Test under Evaluation	Comparative Test	DSe (95% CL)	DSp (95% CL)	Overall Agreement	Κ (+SE) ^1^
U/M RT-rPCR	Cell culture	0.94 (0.80–0.99)	0.99 (0.92–1.00)	97.1%	0.93 (0.10)
	N Uni RT-rPCR	0.94 (0.80–0.99)	0.97 (0.82–1.00)	95.2%	0.90 (0.13)
N Uni RT-rPCR	Cell culture	0.97 (0.85–1.00)	0.97 (0.82–1.00)	96.8%	0.93 (0.13)

^1^ Κ indicates the strength of agreement between two tests, with values of 0.81–0.99 as almost perfect and 1.00 as perfect (Smith 2006).

## Data Availability

All data and machine-readable metadata are publicly available from the USGS ScienceBase digital repository: https://doi.org/10.5066/P963M863; (accessed on 8 July 2022).

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
