# Peer review of "Rapid Diagnostic Test to Detect and Discriminate Infectious Hematopoietic Necrosis Virus (IHNV) Genogroups U and M to Aid Management of Pacific Northwest Salmonid Populations"

_animals, 2022, doi:10.3390/ani12141761_

Round 1

Reviewer 1 Report

A very well designed work. I congratulate the work team. A very important data has been revealed about a subject that is needed and waiting for a solution about IHNV.

” 2.4. According to the numbers given in the diagnostic validation section, the number of samples should be 102. It is suggested to correct 103 to 102 on Line 183 and 260.”

Author Response

” 2.4. According to the numbers given in the diagnostic validation section, the number of samples should be 102. It is suggested to correct 103 to 102 on Line 183 and 260.”

Response: We thank the reviewer for their time and attention to detail.  We have changed the N to 102 in both places.

Reviewer 2 Report

I have few following comments or suggestions. 

Line 105 or 115 – It will be easier for the reader to understand if you can mention somewhere that the forward and reverse primers for both N and U/M assay is same.

Line 260 and 272 – Can you correct or specify which one is right 102 or 103?

Line 255 – Can you add the range of CT values that you got from the wild samples for both N and U/M assay?  

Line 318 – typo ‘farmed’?

Author Response

Line 105 or 115 – It will be easier for the reader to understand if you can mention somewhere that the forward and reverse primers for both N and U/M assay is same.

Response: We thank the reviewer for their time and attention to detail.  The N and U/M assays forward and reverse primers are not the same, as stated on lines 105-106.  I believe this confusion arose because of the nomenclature we used for the F and R primers in Table 1.  We changed Table to 1 to be clearer that these are the F and R for the U/M assays. I edited lines 105-106 to also be clearer on this point.

Line 260 and 272 – Can you correct or specify which one is right 102 or 103?

Response: N = 102.  This has been corrected in the manuscript. 

Line 255 – Can you add the range of CT values that you got from the wild samples for both N and U/M assay?  

Response: Added to lines 259 – 260.

Line 318 – typo ‘farmed’?
Response: Corrected

Reviewer 3 Report

This work presents a PCR-based test to discriminate two genotypes of IHNV, a fish rhabdovirus. The objectives are clearly explained and appear fulfilled, despite some limits in the detection of a particular variant, due to a SNP. The use of a one-step RT-PCR in the whole evaluation of the method would have been better. However, the authors announce promising preliminary results.

I recommend publication after very minor corrections.

L56. A recent reference (2022) may be cited in addition or replacement of Walker 2018.

L.60. The name OIE has recently changed to WOAH, but the change in this text is not mandatory.

L. 103-104. Is there a reason why the probes differ by the 5’ extremity (AGCC for the M probe vs CC for the U probe) ?

L.207. correct “Magnitude”

L. 248-250. Within-run…and Between-run

L. 261. Can you briefly detail the 3 discrepancies ?

L.318. farmed.

L. 346. amplification

Author Response

This work presents a PCR-based test to discriminate two genotypes of IHNV, a fish rhabdovirus. The objectives are clearly explained and appear fulfilled, despite some limits in the detection of a particular variant, due to a SNP. The use of a one-step RT-PCR in the whole evaluation of the method would have been better. However, the authors announce promising preliminary results.

Response:  We appreciate the reviewer's thoughtful comments and attention to detail. Unfortunately, neither of our laboratories use a one-step workflow.  However, we found the one-step master mix gave similar results and we propose that an in-house comparability assessment would be sufficient to adapt the assay (rather than a full validation).

I recommend publication after very minor corrections.

 L56. A recent reference (2022) may be cited in addition or replacement of Walker 2018.

Response: updated the reference

L.60. The name OIE has recently changed to WOAH, but the change in this text is not mandatory.

Response: Thank you for making us aware of the rebranding.  We do write out the full name "World Organization of Animal Health", but we will keep the OIE acronym. This is because we are citing references that use the OIE acronym. 

  1. 103-104. Is there a reason why the probes differ by the 5’ extremity (AGCC for the M probe vs CC for the U probe) ?

Response:  The LNA probes require certain melting temperatures. These were the probes that gave us the right temperature parameters for the assay.  No change to the text.

L.207. correct “Magnitude”

Response: Thank you.  Corrected.

  1. 248-250. Within-run…and Between-run

Response: Thank you.  Corrected.

  1. 261. Can you briefly detail the 3 discrepancies?

Response: We added the following description to lines 262-263: “Two of the discrepant samples tested positive by cell culture and negative by the U/M assay and one sample tested positive by U/M assay (36.7 CT) but negative by cell culture.”

L.318. farmed.

Response: Corrected

L346. amplification

Response: Corrected